# Smelling Peppers and Pout Submitted to Convective Drying: Mathematical Modeling, Thermodynamic Properties and Proximal Composition

**DOI:** 10.3390/foods12112106

**Published:** 2023-05-24

**Authors:** Rodrigo Leite Moura, Rossana Maria Feitosa de Figueirêdo, Alexandre José de Melo Queiroz, Francislaine Suelia dos Santos, Antônio Gilson Barbosa de Lima, Pedro Francisco do Rego Junior, Josivanda Palmeira Gomes, Wilton Pereira da Silva, Yaroslávia Ferreira Paiva, Henrique Valentim Moura, Eugênia Telis de Vilela Silva, Caciana Cavalcanti Costa, Mailson Gonçalves Gregório

**Affiliations:** 1Department of Process Engineering, Federal University of Campina Grande, Campina Grande 58429-900, Brazil; mourarodrigoleite@gmail.com (R.L.M.); antonio.gilson@ufcg.edu.br (A.G.B.d.L.); yaroslaviapaiva@gmail.com (Y.F.P.); 2Department of Agricultural Engineering, Federal University of Campina Grande, Campina Grande 58429-900, Brazil; rossanamff@gmail.com (R.M.F.d.F.); alexandrejmq@gmail.com (A.J.d.M.Q.); josivanda@gmail.com (J.P.G.); wiltonps@uol.com.br (W.P.d.S.); valentim_henrique@hotmail.com (H.V.M.); eugenia_telys@hotmail.com (E.T.d.V.S.); caciana.cavalcanti@professor.ufcg.edu.br (C.C.C.); gregoriomailson@gmail.com (M.G.G.); 3Federal Institute of Education, Science and Technology of Ceará, Campus Quixadá, Quixadá 62930-000, Brazil; pfrj23@gmail.com

**Keywords:** *Capsicum chinense*, *Capsicum chinense* Jacq., dehydration, effective diffusivity, physical-chemical composition

## Abstract

Pepper (*Capsicum* spp.) is among the oldest and most cultivated crops on the planet. Its fruits are widely used as natural condiments in the food industry for their color, flavor, and pungency properties. Peppers have abundant production; on the other hand, their fruits are perishable, deteriorating within a few days after harvesting. Therefore, they need adequate conservation methods to increase their useful life. This study aimed to mathematically model the drying kinetics of smelling peppers (*Capsicum chinense*) and pout peppers (*Capsicum chinense* Jacq.) to obtain the thermodynamic properties involved in the process and to determine the influence of drying on the proximal composition of these peppers. Whole peppers, containing the seeds, were dried in an oven with forced air circulation, at temperatures of 50, 60, 70, and 80 °C, with an air speed of 1.0 m/s. Ten models were adjusted to the experimental data, but the Midilli model was the one that provided the best values of coefficient of determination and lowest values of the mean squared deviation and chi-square value in most of the temperatures under study. The effective diffusivities were well represented by an Arrhenius equation, appearing in the order of 10^−10^ m^2^·s^−1^ for both materials under study, since the activation energy of the smelling pepper was 31.01 kJ·mol^−1^ and was 30.11 kJ·mol^−1^ in the pout pepper, respectively. Thermodynamic properties in both processes of drying the peppers pointed to a non-spontaneous process, with positive values of enthalpy and Gibbs free energy and negative values of entropy. Regarding the influence of drying on the proximal composition, it was observed that, with the increase in temperature, there was a decrease in the water content and the concentration of macronutrients (lipids, proteins, and carbohydrates), providing an increase in the energy value. The powders obtained in the study were presented as an alternative for the technological and industrial use of peppers, favoring obtaining a new condiment, rich in bioactives, providing the market with a new option of powdered product that can be consumed directly and even adopted by the industry as a raw material in the preparation of mixed seasonings and in the formulation of various food products.

## 1. Introduction

Pepper (*Capsicum* spp.) is among the oldest and most cultivated crops on the planet [1,2]. The genus *Capsicum* belongs to the Solanaceae family, with approximately 31 types, among which the five main cultivated ones are *C. annuum*, *C. baccatum*, *C. chinense*, *C. frutescens*, and *C. pubescens*. Among these, *C. frutescens* and *C. chinense* are the most widely cultivated [3]. *Capsicum chinense* is considered the most Brazilian of the species, with high genetic diversity in the Amazon Basin, and it is very popular throughout the tropical region. Its diversity is more evident in fruits in terms of size, color, shape, aroma, and degree of pungency [4,5]. *C. chinense* has several cultivars, and the best known are smelling, Cumari do Pará, Murupi, habanero, bode, and pout [5,6]. Peppers are widely used as natural condiments in the food industry for their color, flavor, and pungency properties. When peppers are dehydrated, they are regularly used in various types of products such as pizza, food mixes, salad dressing, and instant soups [7]. Peppers have abundant production; on the other hand, their fruits are perishable, deteriorating within a few days after harvesting. Therefore, they need adequate conservation methods to increase their shelf life, minimizing loss or damage to their bioactive and nutritional compounds [8].

Drying is a very important operation in the industry that is traditionally adopted in the preservation of perishable products, helping to maintain quality and stability by reducing humidity and water activity. In this perspective, convective drying is shown to be a widely used and suitable technology for food, perishable vegetables, and by-products that is generally applied for the dehydration of agro-industrial products, bringing the advantages of reducing the stored volume and increasing the shelf life [9]. Maurya et al. [7] listed different drying methods adopted in the dehydration of agricultural and food products, such as sun drying, hot air drying, microwaves, and freeze-drying. Conventional hot air drying has become a popular method due to its relatively short time and uniform heating. Studies of drying kinetics combined with mathematical modeling and the evaluation of thermodynamic properties deserve the interest of researchers in the area of the most diverse products, considering the diversity of biological structures involved in heating and mass transfer and the effects observed in each material [8].

By studying the effective diffusivity, it is possible to obtain information about how the structure of the food affects the mass transfer rate. Different foods have different compositions and structures, which can affect their moisture diffusing ability. Determining this parameter allows for the development of more accurate mathematical models to predict the drying rate and the time required to reach a certain desired residual moisture level [10]. Additionally, from the study of the thermodynamic properties of the process, the following are possible: to determine the amount of heat necessary for the evaporation of water from the enthalpy; to predict the drying rate and control the quality of the final product and how to avoid problems such as the formation of unwanted crusts and agglomerates from entropy; and to determine the viability of the process through Gibbs free energy, making it possible to design and control the drying process, ensuring that it occurs efficiently and economically [11].

The powders obtained in the study are presented as an alternative for the technological and industrial use of peppers, which favor obtaining a new condiment, rich in bioactives, providing the market with a new option of powdered product that can be consumed directly and even adopted by the industry as a raw material in the preparation of mixed seasonings and in the formulation of various food products. Therefore, given the above, the present work aimed to mathematically model the drying kinetics of smelling peppers (*Capsicum chinense*) and pout peppers (*Capsicum chinense* Jacq.), to obtain the thermodynamic properties involved in the process and to determine the influence of drying on the proximal composition of these peppers.

## 2. Materials and Methods

### 2.1. Raw Material and Processing

The raw materials used were whole pepper fruits (containing the seeds) of smelling and pout varieties, which were purchased at the Empresa Paraibana de Abastecimento e Serviços Agrícolas (EMPASA) in the city of Campina Grande, Paraíba, Brazil. The smelling peppers were collected at the green maturation stage (green colored fruits) and the pout peppers at the mature maturation stage (red colored fruits). The fruits were then transported to the laboratory, where they were manually selected in order to eliminate those that presented physical damage and/or different coloration from the defined maturation stage. After selection, they were washed in running water and then immersed in a 100 ppm sodium hypochlorite solution for 20 min for sanitization, and then they were rinsed and placed in plastic trays lined with paper towels, exposed to the environment on a laboratory bench, to eliminate excess surface water. Afterwards, the peppers were sliced/cut in half (keeping the seeds) and arranged in stainless steel trays.

### 2.2. Drying Kinetics

For the drying kinetics experiment, in triplicate, the samples were weighed in stain-less-steel trays and were dried in an oven with forced air circulation (Fanem, model 320, Guarulhos, São Paulo, Brazil) at temperatures of 50, 60, 70, and 80 °C, with relative humidities of 67, 64, 62, and 60%, respectively, and air velocity of 1.0 m·s^−1^ until the equilibrium water content was reached. Parameters were determined according to the excellent results obtained in previous research with similar raw materials [12,13].

The reduction of water content during drying was monitored by the gravimetric method (mass loss), weighing the samples at time intervals of 5, 10, 15, 20, 30, 40, 50, and 60 min, using a semi-analytical balance (Marte, model AS5500C, Santa Rita do Sapucaí, Minas Gerais, Brazil). When the samples reached constant mass, the equilibrium water content was determined, as well as the dry mass in an oven at 105 °C, according to the methodology AOAC [14]. The dried samples were removed from the trays with the aid of a stainless steel spatula and were ground in a knife mill (Marconi, model TE 340, Piracicaba, São Paulo, Brazil) to obtain the powders. With the experimental data of the drying kinetics, the water content ratios of the samples were calculated, according to Equation (1):(1)$$\mathrm{MR}=\frac{X\;-{\;X}_{e}}{{X\;}_{i}-{\;X}_{e}}$$
where MR is the sample water content ratio (dimensionless); X is the water content of the sample at a given drying time (d.b.); X_i_ is the initial water content of the sample (d.b.); X_e_ is the equilibrium water content of the sample (d.b.).

Table 1 presents the 10 mathematical models that were adjusted to the experimental data on the drying kinetics of peppers. To adjust the different models, the computer program Statistica 7.7^®^ was used with a non-linear regression analysis, using the Quasi-Newton method.

To evaluate the quality of the adjustments of the different mathematical models, the following criteria were used: the magnitude of the determination coefficient (R^2^), the mean squared deviation (MSD), and the chi-square value (χ^2^), according to Equations (12), (13) and (14), respectively:(12)$$R^{2}=1\;-\left(\frac{\sum_{i=1}^{n}{{(\mathrm{MR}}_{\mathrm{pred},i}-{\;\mathrm{MR}}_{\exp,i})}^{2}}{\sum_{i=1}^{n}{\left({\mathrm{MR}}_{\exp,i}-{\;\mathrm{MR}}_{\mathrm{pred},i}\right)}^{2}}\right)$$
(13)$$\mathrm{MSD}={\left[\frac{1}{n}\sum_{i=1}^{n}{{(\mathrm{MR}}_{\mathrm{pred},i}-{\;\mathrm{MR}}_{\exp,i})}^{2}\right]}^{\frac{1}{2}}$$
(14)$$\chi^{2}=\frac{1}{n-N}\sum_{i=1}^{n}{{(\mathrm{MR}}_{\exp,i}-{\;\mathrm{MR}}_{\mathrm{pred},i})}^{2}$$
where R^2^ is the coefficient of determination; MSD is the mean squared deviation; χ^2^—chi-square; MR_pred,i_ is the water content ratio predicted by the model; MR_exp,i_ is the experimental water content ratio; n is the number of observations; N is the number of model constants.

### 2.3. Effective Diffusivity

Effective moisture diffusivity was determined by fitting the mathematical model of liquid diffusion with the approximation of nine terms (Equation (15)) to the experimental data of the drying kinetics of the peppers at different temperatures, considering uniform initial moisture distribution, constant diffusivity, and negligible thermal resistance and volumetric shrinkage. This model is the analytical solution for Fick’s second law, considering the geometry approximately as a flat plate, distributed along the tray for drying (area >> thickness) [21].
(15)$$\mathrm{MR}=\frac{X\;-{\;X}_{e}}{X_{I}-{\;X}_{e}}=\frac{9}{\pi^{2}}\sum_{n=0}^{\infty }\frac{1}{{\left(2n+1\right)}^{2}}\exp\left[-{\left(2n+1\right)}^{2}\pi^{2}D_{\mathrm{eff}}\frac{t}{{4L}^{2}}\right]\;$$
where D_ef_ is the effective diffusivity (m^2^·s^−1^); n is the number of terms in the equation; L is the characteristic dimension, half thickness (m); t is time (s).

The relationship between effective moisture diffusivity and drying temperature was described by an Arrhenius-type equation (Equation (16)) [22]:(16)$$D_{\mathrm{eff}}{=D}_{0}\left(-\frac{E_{a}}{\mathrm{RT}}\right)$$
where D_eff_ is the pre-exponential factor (m^2^·s^−1^); E_a_ is the activation energy (kJ·mol^−1^); R is the universal gas constant (0.008314 kJ·mol^−1^·K^−1^); T is the absolute temperature (K).

Arrhenius-type equation parameters were obtained by linearizing Equation (16) and applying the natural logarithm, according to Equation (17) [23]:(17)$${\mathrm{LnD}}_{\mathrm{eff}}{=\mathrm{LnD}}_{0}\left(-\frac{E_{a}}{\mathrm{RT}}\right)$$
where LnD_0_ is the logarithmic of the pre-exponential factor (m^2^·s^−1^), E_a_ is the activation energy (J·mol^−1^), R is the universal gas constant (8.314 J·mol^−1^·K^−1^), and T is the absolute temperature (K).

### 2.4. Thermodynamic Properties

The thermodynamic properties (enthalpy, entropy, and Gibbs free energy) of the seed drying process at temperatures of 50, 60, 70, and 80 °C were quantified using the method described by Wanderley et al. [20], according to Equations (18)–(20):(18)$${\Delta H=E}_{a}-\;\mathrm{RT}$$
(19)$$\Delta S=R\left[\ln\left(D_{\mathrm{ef}}{}_{0}\right)-\;\ln\left(\frac{k_{B}}{h_{p}}\right)-\;\ln\left(T\right)\right]$$
(20)ΔG=ΔH −TΔS
where ΔH is the specific enthalpy (J·mol^−1^); ΔS is the specific entropy (J·mol^−1^·K^−1^); ΔG is the Gibbs free energy (J·mol^−1^); E_a_ is the activation energy (J·mol^−1^); R is the universal gas constant (8.314 J·mol^−1^·K^−1^); D_ef0_ is the pre-exponential factor (m^2^·s^−1^); K_B_ is the Boltzmann constant (1.38 × 10^−23^ J·K^−1^); h_p_ is Planck’s constant (6.626 × 10^−34^ J·s^−1^); T is the absolute temperature (K).

### 2.5. Proximal Composition of in Natura and Dried Peppers

For peppers in natura and dried, the following analyses were carried out in triplicate, according to the analytical procedures of the AOAC [14]: water content, by the gravimetric method in an oven (New Lab, model NL 82-27, Piracicaba, São Paulo, Brazil) at 105 °C, to constant mass; ash content, by muffle (Forlabo, modelo BioFM 6.7 L, Diadema, São Paulo, Brazil) incineration at 550 °C with results expressed in percentage (p/p); protein content based on nitrogen determination by the Kjeldahl digestion process, followed by distillation and titration.

The lipid content was determined by extraction with a cold mixture of solvents, according to the method described by Bligh and Dyer [24]. The total carbohydrates were determined by the difference of the other constituents (proteins, lipids, ash, and water content), obtaining the Nifext fraction, according to RDC n° 360 [25]. The energy value was determined by the means of the Atwater conversion factors: 4 kcal/g for carbohydrates and proteins and 9 kcal/g for lipids [26].

## 3. Results and Discussion

### 3.1. Drying Kinetics

Table 2 presents the reduction in drying time, as well as in the water content on a wet and dry basis, as there was an increase in temperature. For smelling peppers, in the variation from 50 to 80 °C, there was a reduction of more than 300% in the drying time, as well as a reduction of 41.52 and 44.04% for the water content on a wet and dry basis, respectively.

Still, according to Table 2, for pout peppers, in the variation from 50 to 80 °C, there was a reduction of more than 200% in the drying time, as well as a reduction of 30.79 and 40.04% for water content on a wet and dry basis, respectively. A reduction in the equilibrium water content with increasing drying air temperature has also been reported for fruits of Cumari-do-Pará pepper [27], Moroccan pepper [28], Cabacinha pepper [29], red pepper [30], green pepper [31], Cabacinha pepper seeds [19], and Kampot pepper [32]. This result indicates that the increase in temperature acts in the intensification of the drying process since higher temperatures can contribute to the improvement of the transfer of heat between the drying air and the samples, promoting a greater loss of water. Kumar et al. [33] pointed out that this intensification is related to the increase in the internal partial vapor pressure due to the increase in temperature, which amplifies the difference between the vapor pressure of the drying air and the product, allowing the water to be eliminated more easily and quickly.

Table 3 and Table 4 show the parameters, the coefficients of determination (R^2^), the mean squared deviations (MSD), and the chi-square (χ^2^) of the mathematical models adjusted to the experimental data of the drying of smelling peppers and pout peppers at temperatures from 50 to 80 °C. The coefficients of determination (R^2^) showed values greater than 0.98 in all models and at all temperatures evaluated. For smelling peppers, at a temperature of 50 °C, the Logarithmic model resulted in the highest coefficient of determination (R^2^ = 0.9997) and the lowest values for MSD and χ^2^. At the other temperatures (60, 70, and 80 °C), the Midilli model provided the highest values of the coefficient of determination (R^2^ > 0.9967) and the lowest values for MSD and χ^2^, resulting in very close adjustment parameters.

For pout peppers, at a temperature of 50 °C, the Two Term model showed the highest coefficient of determination (R^2^ = 0.9997) and the lowest values for MSD and χ^2^. At temperatures of 60 and 70 °C, the Verna model obtained the highest coefficient of determination values (R^2^ > 0.9987) and the lowest values for MSD and χ^2^. At a temperature of 80 °C, the Midilli model resulted in the highest coefficient of determination (R^2^ = 0.9995) and the lowest values for MSD and χ^2^.

Comparing the magnitude of the R^2^, for the smelling peppers, it was observed that the Logarithmic, Midilli, Page, and Verna models presented values of R^2^ ≥ 0.9900 in all temperatures, while the other models presented an R^2^ from 0.9800, which indicates a good fit of the models to the experimental data. For the pout peppers, when comparing the magnitude of R^2^, it was observed that the Two Term, Henderson and Pabis, Henderson and Pabis Modified, Logarithmic, Logistic, Midilli, Page, and Verna models obtained R^2^ values ≥ 0.9900 in all temperatures, while the Newton and Thompson models presented an R^2^ from 0.9800, also indicating a good fit of the models to the experimental drying data.

Silva et al. [29] studied the drying kinetics of Cabacinha pepper fruits at temperatures of 60, 70, 80, 90, and 100 °C, reporting that the Midilli model was the one that presented the best adjustment to the experimental drying data, with an R^2^ > 0.9990. Tekin and Baslar [34] investigated the effect of ultrasound-assisted vacuum drying on red peppers (*Capsicum annuum* L.) at temperatures of 45, 55, 65, and 75 °C, noting that the Logarithmic model showed a better fit, with R^2^ > 0.9950. Souza et al. [35] evaluated the drying kinetics of pequi mesocarp, reporting that, among the analyzed models, Midilli presented the most adequate fit to describe the drying curves at temperatures of 40, 50, 60, and 70 °C, with an R^2^ > 0.9979 and the lowest values for mean relative error and estimated mean error. Midilli et al. [19] developed this model through experiments with mushrooms, pollen, and pistachios dried in laboratory conditions, under direct sunlight and in a system with a combination of sunlight and forced air circulation, in order to describe the drying kinetics in a single layer. They compared data statistically with other commonly used empirical or semi-empirical models.

Still, according to Table 3 and Table 4, it can be verified for the Midilli model that the coefficient did not increase as the drying temperature increased. The other coefficients, a, k, and b, did not express a clear trend as a function of temperature. The variations of the a, n, and b coefficients, unlike the k coefficient, were more associated with the mathematical modeling of the adjustments than with the drying phenomenon, since Midilli is a semi-empirical model [36]. Tekin and Baslar [34] observed the same behavior of the coefficients involved in modeling the drying of red peppers.

Figure 1 shows the drying curves for smelling peppers (Figure 1a) and pout peppers (Figure 1b) adjusted by the Midilli model at temperatures from 50 to 80 °C. In both varieties, the increase in drying temperature resulted in an increase in the water evaporation rate from the samples. For smelling peppers, the curves showed different behavior for different temperatures. In pout peppers, at temperatures of 70 and 80 °C, the curves showed similar behavior. There is also a low dispersion of the experimental values in relation to the adjustment curves, which was a result of the high values of R^2^ and low MSD and χ^2^.

### 3.2. Effective Diffusivity

Table 5 shows the effective diffusivities (D_eff_) obtained in the drying of smelling and pout peppers at the studied temperatures. The diffusivity values increased with the increase in the temperature of the drying air by approximately 264 and 247% for smelling and pout peppers, respectively, among the evaluated temperatures, including all values in the range frequently reported for foodstuffs, which is from 10^−9^ to 10^−11^ m^2^ s^−1^ [32].

The effective diffusivity is directly influenced by the increase in temperature, since this change modifies the removal of water from the sample. Gandolfi et al. [31], evaluating the drying of green peppers at two temperatures (60 and 75 °C) and two air speeds (1.5 and 3.0 m s^−1^), reported D_eff_ values above those usually reported for food products, ranging from 4.63 × 10^−8^ to 7.98 × 10^−8^, indicating that the increase in temperature had a greater influence than the air velocity on the increase in diffusivity. Silva et al. [29] indicated D_eff_ values ranging from 4.07 × 10^−9^ to 21.42 × 10^−9^ in the drying of Cabacinha pepper fruits at temperatures from 60 to 100 °C.

Differences in diffusivity values can be caused by the chemical composition, lipid content, initial amount of water, and especially by the structure, shape, and size of each material. In the migration of water from the interior to the surface of the product, the physical and chemical properties are influenced by the composition, depending on a greater or lesser affinity, which is also related to the structural nature of each raw material [37].

In Figure 2, there is a representation of the effective diffusivities (D_eff_), in the form of ‘ln D_eff_’, as a function of the inverse of the absolute temperature (1/T), referring to the drying of the peppers under study. It was observed that the dependence was satisfactorily described by the Arrhenius-type equation, with values revealing linear behavior and R^2^ values above 0.9770.

Table 6 shows the values of activation energy for smelling and pout peppers quantified from the Arrhenius-type equation (Figure 2).

Activation energy refers to the amount of energy required to stimulate the water diffusion process. Therefore, the lower the activation energy, the greater the effective diffusivity of the water contained in the material; consequently, the energy required to transform liquid free water into steam will be lower [38]. This energy must be overcome to allow the migration of water molecules from the inside to the outside of the product and varies according to structure and chemical composition. It was noted that the drying process of smelling and pout peppers showed very close activation energies, with a slightly higher value for Smelling peppers, with values that fall within the range of 12.7 to 110 kJ·mol^−1^, often reported for foodstuffs [39].

Higher values than the present study were reported by Kumar and Shrivastava [40] for green peppers (43.38 kJ·mol^−1^); Silva et al. [29] for Cabacinha pepper (36.09 kJ·mol^−1^); Vega et al. [41] for red peppers (39.70 kJ·mol^−1^); and Kaleemullah and Kailappan [42] for red peppers (37.76 kJ·mol^−1^). Onwude et al. [43] carried out a comprehensive review study on the thin-layer drying of fruits and vegetables and found, in the researched bibliography, that 90% of the values for activation energy were in the range of 14.42 to 43.26 kJ·mol^−1^.

### 3.3. Thermodynamic Properties

Table 7 shows the thermodynamic properties of the drying process of smelling and pout peppers, with the respective enthalpy, entropy, and Gibbs free energy for the evaluated temperatures. The enthalpy (ΔH) decreased with increasing drying temperature, which indicates that a smaller amount of energy was required for drying to occur at higher temperatures. Silva et al. [29] reported equivalent behavior, with ΔH values ranging from 33.3233 to 32.9908 kJ·mol^−1^ for the enthalpy in drying Cabacinha peppers at temperatures from 60 to 100 °C. During the drying process, the enthalpy is related to the energy required to remove the water bound to the product. In our study, the pout pepper had a lower ΔH than the smelling pepper, indicating a lower amount of energy needed to dry them, corroborating their initial water content (Table 8).

Entropy behaved similarly to enthalpy, showing a slight decrease with increasing temperature and with higher values for smelling pepper. Entropy is a thermodynamic property associated with the degree of disorder of the system, and when it reduces with increasing temperature, it indicates that there is an increase in the order of the system [11].

The Gibbs free energy (ΔG) was positive and increased with temperature, indicating an endergonic reaction, which requires the addition of energy from the environment involved in the drying of the product. This energy is related to the work required to make the sorption sites available, and this behavior was expected since the desorption process was not spontaneous. Values close to ΔG were determined by Rodovalho et al. [44], who evaluated the drying kinetics of goat pepper grains at temperatures of 30, 35, and 40 °C, reporting values for ΔG of 113.15, 135.10, and 137.05 kJ·mol^−1^, respectively.

### 3.4. Proximal Composition of in Natura and Dried Peppers

Table 8 presents the average values that resulted from the proximal composition of the peppers in natura and dried at temperatures of 50, 60, 70, and 80 °C. Drying promoted a reduction in water content and an increase in lipid concentration, influencing the energy value. According to Leite et al. [10], the degradation of nutrients due to heat is often compensated by the concentration in view of the reduction in water content. However, the volatilization of thermolabile principles or the conversion into other compounds can cause changes in composition, even causing a decrease in nutritional values, which explains the variations observed for carbohydrates, ashes, and proteins.

For the parameters evaluated in the proximal composition, statistically significant differences were observed when comparing the two varieties of peppers and the applied treatments. Peppers in natura showed high water content (above 80%) and the increase in drying air temperature reduced the water content to average values between 6 and 8%. The loss of water observed with the increase in drying temperature promoted less water availability for the activity of microorganisms and enzymes, which increases the shelf life of the product. Vega-Gálvez et al. [45] studied the effect of drying air temperature on the physicochemical properties of red pepper, reporting a water content of 89.40% for fresh samples. A maximum water content value of 10.63% was observed at 70 °C; however, lower water contents were observed at 50 °C (8.75%) and 60 °C (9.01%), as well as at 80 °C (8.72%) and 90 °C (8.88%) due to long drying times and high temperatures.

In the determination of lipids, there is a tendency for their concentration to increase with an increase in temperature. When in natura there was no significant difference between the analyzed peppers and, after drying, the pout pepper had a higher lipid content with statistically significant differences between the analyzed varieties. Reis et al. [46] found the following percentages for Cumari do Pará peppers in different treatments: 0.52% (in natura, 45 and 55 °C) and 0.49% (65 °C), lower values compared to the present study, which can be explained by the probable absence of seeds, since whole peppers were used in this study.

Proteins showed slight changes. For in natura smelling peppers compared to dried peppers, a slight decrease was observed. In pout peppers, a subtle addition to the dried ones could be seen in comparison with the in natura treatment. Pinar et al. [47] evaluated two pepper cultivars in different drying methods, reporting 7.97 and 9.45% of crude protein for the Bozok cultivar at temperatures of 60 and 80 °C and for the Pinar cultivar, 7.68% (60 °C) and 10.68% (80 °C) crude protein. It was noted that, at the highest temperature, there was a better preservation of the protein content, and similar behavior occurred in the present study, where the protein value was higher at 80 °C than at 50 °C. It was noticed that, in both varieties, the protein content at the temperature of 80 °C did not differ significantly in relation to in natura treatments, confirming the non-degradation of the proteins. This happens because higher temperatures reduce enzymatic activity, thus contributing to the prevention of losses.

The ashes also showed few changes that were mostly non-significant, both in comparison among varieties and among drying temperatures and in natura samples. Faustino, Barroca, and Guiné [48], in a study on the drying and characterization of green peppers, obtained percentages of 5.14% (in natura), 6.78% (30 °C), and 6.42% (70 °C), not showing significant differences between fresh and dried samples. Similar behavior was verified for smelling peppers, where the mean value obtained at 80 °C did not differ significantly from the sample in natura.

Carbohydrates revealed values between 71 and 77% for peppers in natura. For smelling peppers dried at 50 °C, there was a significant increase in the average value, with a tendency to decrease for the other temperatures, whereas the pout pepper showed a tendency for a decrease for all temperatures in comparison with the in natura treatment. Faustino, Barroca, and Guinea [48], when evaluating green peppers, mentioned the following percentages for total carbohydrates: 85.27% (in natura), 4.36% (30 °C), and 4.58% (70 °C), revealing a significant loss, where the drying operation induced a reduction of 95% of these. However, in the present study, slight variations were observed, where for smelling peppers, the average value obtained at 80 °C did not differ statistically from the in natura treatment. There was no significant difference between the two varieties evaluated at this temperature. This behavior suggests the existence of some protective mechanism or component activated during the drying of these peppers.

The energy value in kilocalories (kcal 100^−1^ g^−1^ d.b.) for both varieties under study, in the in natura treatment, there were very close values with no significant difference. It was possible to notice that the elevation of the drying temperature promoted a tendency to increase the energy value, since the reduction of the water content tends to concentrate the macronutrients, providing the increase of the caloric value. Vega-Gálvez et al. [45] determined the proximal composition of fresh red pepper, quoting an energy value of 272.17 kcal 100^−1^ g^−1^. From the composition carried out by Faustino, Barroca, and Guiné [48] for green peppers in natura, it is possible to indicate an energy value of 471.50 kcal 100^−1^ g^−1^, placing the values of the present study within the range mentioned by the authors.

## 4. Conclusions

The applied models presented good adjustments in the description of the drying kinetics of the smelling peppers and pout peppers, the Midilli model was the one that provided the best values of coefficient of determination and lowest values of the mean squared deviation and the chi-square value in most of the temperatures under study. Effective diffusivity accompanied the increase in drying temperature, with the relationship satisfactorily adjusted by an Arrhenius-type equation. The thermodynamic properties were influenced by increasing the drying temperature, with a reduction in enthalpy and an increase in the Gibbs free energy.

The peppers showed considerable amounts of proteins, lipids, and carbohydrates, with the drying process influencing the proximal composition and energy value. Then, it was found that the powders obtained in the study were presented as an alternative for the technological and industrial use of peppers that favored obtaining a new condiment, rich in bioactives, providing the market with a new option of powdered product that can be consumed directly and even adopted by the industry as a raw material in the preparation of mixed seasonings and in the formulation of various food products.

## 5. Patents

Patents resulting from the work reported in this manuscript are as follows:-Production of the biquinho-type red pepper (*Capsicum chinense* Jacq.) powder—BR 10 2021 014 391 6;-Green pepper (*Capsicum chinense*) powder—BR 10 2021 014 319 3.

## Figures and Tables

**Figure 1 foods-12-02106-f001:**
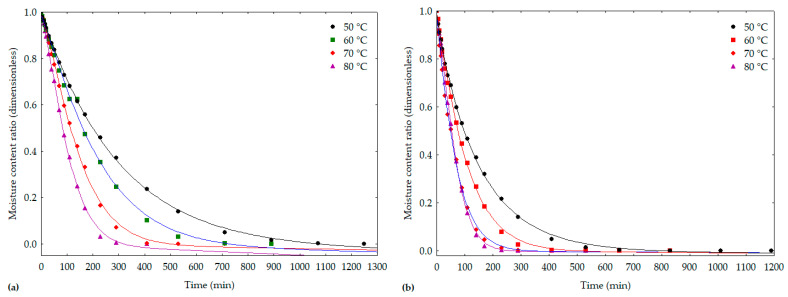
Curves of drying kinetics at temperatures from 50 to 80 °C and adjusted by the Midilli model for peppers: (**a**) smelling; (**b**) pout.

**Figure 2 foods-12-02106-f002:**
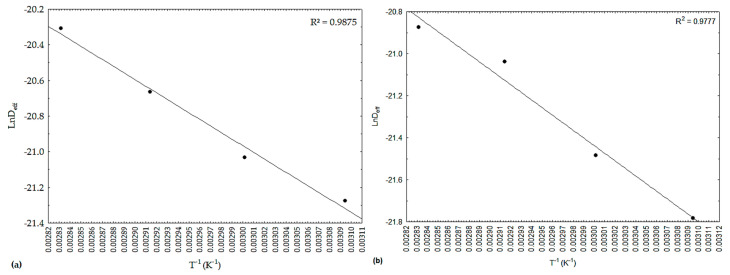
Representation of effective diffusivity by the Arrhenius-type equation for drying peppers: (**a**) smelling; (**b**) pout.

**Table 1 foods-12-02106-t001:** Mathematical models adjusted to the drying kinetic data of peppers.

Equation	Model Designation	Mathematical Model	References
(2)	Two terms	MR = a·exp(−k_0_·t) + b exp(−k_1_·t)	[15]
(3)	Henderson and Pabis	MR = a·exp(−k·t)	[16]
(4)	Henderson and Pabis modified	MR = a·exp(−k·t) + b·exp(−k_0_·t) + c·exp(−k_1_·t)	[17]
(5)	Logarithmic	MR = a·exp(−k·t) + c	[8]
(6)	Logistic	MR = a_0_/(1a·exp(k·t))	[18]
(7)	Midilli	MR = a·exp(−k·t^n^) + b.t	[19]
(8)	Newton	MR = exp(−k·t)	[10]
(9)	Page	MR = exp(−k·t^n^)	[12]
(10)	Thompson	$\mathrm{MR}=\exp\left(\frac{-a\;-{\;(a}^{2}{+4\mathrm{bt})}^{0.5}}{2b}\right)$	[13]
(11)	Verna	MR = a·exp(−k·t) + (1 − a) exp(−k_2_·t)	[20]

MR is the water content ratio, dimensionless; a, b, c, k, n, and q are the model parameters; t is the drying time, min.

**Table 2 foods-12-02106-t002:** Mean values of drying times and the water content of smelling peppers and pout peppers dried at different temperatures.

Pepper	Temperature (°C)	Drying Time (min)	Water Content (% w.b.)	Water Content (% d.b.)
Smelling	50	1250 ± 0.00	6.34 ± 0.24	6.77 ± 0.27
	60	890 ± 0.00	7.33 ± 0.49	7.91 ± 0.57
	70	710 ± 0.00	6.47 ± 0.26	6.92 ± 0.29
	80	410 ± 0.00	4.48 ± 0.19	4.70 ± 0.23
Pout	50	1190 ± 0.00	5.65 ± 0.10	6.33 ± 0.14
	60	830 ± 0.00	5.02 ± 0.27	5.29 ± 0.30
	70	650 ± 0.00	4.80 ± 0.22	5.04 ± 0.24
	80	530 ± 0.00	4.32 ± 0.12	4.52 ± 0.19

**Table 3 foods-12-02106-t003:** Parameters, determination coefficients (R^2^), and mean squared desviations (MSD) and chi-square (χ^2^) of the mathematical models adjusted to the drying kinetic data of smelling peppers at temperatures of 50, 60, 70, and 80 °C.

Model		Parameters										
Two Term	T (°C)	a		k_0_		b		k_1_		R^2^	MSD	χ^2^
	50	0.5428		0.0035		0.4583		0.0035		0.9989	0.0118	0.0002
	60	0.5102		0.0046		0.5081		0.0046		0.9927	0.0302	0.0012
	70	0.5237		0.0068		0.5237		0.0068		0.9915	0.0339	0.0015
	80	0.5318		0.0098		0.5318		0.0098		0.9876	0.0403	0.0022
Henderson and Pabis	T (°C)	a				k				R^2^	MSD	χ^2^
	50	1.0012				0.0035				0.9989	0.0118	0.0002
	60	1.0183				0.0046				0.9927	0.0302	0.0010
	70	1.0475				0.0068				0.9915	0.0339	0.0013
	80	1.0588				0.0097				0.9876	0.0402	0.0018
Henderson and Pabis Modified	T (°C)	a	k		b	k_0_	c		k_1_	R^2^	MSD	χ^2^
	50	0.3365	0.0035		0.3320	0.0035	0.3326		0.0035	0.9989	0.0118	0.0002
	60	0.3338	0.0046		0.3347	0.0046	0.3498		0.0046	0.9927	0.0302	0.0013
	70	0.3490	0.0068		0.3490	0.0068	0.3490		0.0068	0.9915	0.0339	0.0017
	80	0.3529	0.0097		0.3529	0.0097	0.3529		0.0097	0.9876	0.0402	0.0026
Logarithmic	T (°C)	a			k		c			R^2^	MSD	χ^2^
	50	1.0268			0.0033		−0.0305			0.9997	0.0064	0.00005
	60	1.0753			0.0040		−0.0656			0.9955	0.0236	0.00066
	70	1.1000			0.0061		−0.0611			0.9946	0.0272	0.00089
	80	1.1442			0.0080		−0.0999			0.9934	0.0294	0.00106
Logistic	T (°C)	a_0_			a		k			R^2^	MSD	χ^2^
	50	0.0415			0.0415		0.0035			0.9989	0.0118	0.0002
	60	0.0714			0.0701		0.0046			0.9927	00302	0.0011
	70	0.0766			0.0732		0.0068			0.9915	0.0339	0.0014
	80	0.1837			0.1735		0.0097			0.9876	0.0402	0.0020
Midilli	T (°C)	a		k		n		b		R^2^	MSD	χ^2^
	50	0.9938		0.0031		1.0200		−0.00002		0.9996	0.0070	0.0001
	60	0.9839		0.0018		1.1691		−0.00003		0.9967	0.0176	0.0005
	70	0.9923		0.0015		1.2874		−0.00002		0.9990	0.0117	0.0002
	80	0.9895		0.0018		1.3451		−0.00005		0.9992	0.0106	0.0001
Newton	T (°C)	k								R^2^	MSD	χ^2^
	50	0.0035								0.9989	0.0119	0.0001
	60	0.0045								0.9919	0.0318	0.0011
	70	0.0064								0.9871	0.0419	0.0019
	80	0.0089								0.9810	0.0499	0.0027
Page	T (°C)	k				n				R^2^	MSD	χ^2^
	50	0.0030				1.0309				0.9991	0.0106	0.0001
	60	0.0021				1.1445				0.9959	0.0227	0.0006
	70	0.0017				1.2758				0.9987	0.0131	0.0002
	80	0.0020				1.3329				0.9987	0.0131	0.0002
Thompson	T (°C)	a				b				R^2^	MSD	χ^2^
	50	−8206.30				5.3873				0.9989	0.0119	0.0002
	60	−4878.02				4.6680				0.9919	0.0318	0.0011
	70	−2551.27				4.0268				0.9871	0.0419	0.0020
	80	−4143.82				6.0725				0.9809	0.0499	0.0028
Verna	T (°C)	a			k		k_1_			R^2^	MSD	χ^2^
	50	−0.0019			0.1944		0.0035			0.9989	0.0118	0.0002
	60	−0.1863			0.0186		0.0055			0.9952	0.0245	0.0007
	70	−7.1194			0.0130		0.0118			0.9987	0.0133	0.0002
	80	−14.5638			0.0184		0.0175			0.9984	0.0143	0.0003

**Table 4 foods-12-02106-t004:** Parameters, coefficients of determination (R^2^), and mean squared deviations (MSD) and chi-square (χ^2^) of the mathematical models adjusted to data on the drying kinetics of pout peppers at temperatures of 50, 60, 70, and 80 °C.

Model		Parameters									
Two Term	T (°C)	a		k_0_		b	k1		R^2^	MSD	χ^2^
	50	0.0453		0.1071		0.9538	0.0065		0.9997	0.0063	0.00005
	60	0.5715		0.0096		0.4427	0.0096		0.9984	0.0147	0.00028
	70	0.5017		0.0149		0.5017	0.0148		0.9974	0.0183	0.00043
	80	0.5294		0.0152		0.5294	0.0152		0.9912	0.0356	0.00165
Henderson and Pabis	T (°C)	a				k			R^2^	MSD	χ^2^
	50	0.9742				0.0067			0.9993	0.0098	0.0001
	60	1.0141				0.0096			0.9984	0.0147	0.0002
	70	1.0028				0.0148			0.9974	0.0183	0.0004
	80	1.0587				0.0152			0.9912	0.0356	0.0014
Henderson and Pabis Modified	T (°C)	a	k		b	k_0_	c	k_1_	R^2^	MSD	χ^2^
	50	0.3246	0.0062		0.3246	0.0062	0.3263	0.0081	0.9993	0.0098	0.0001
	60	0.3382	0.0096		0.3382	0.0096	0.3378	0.0096	0.9984	0.0147	0.0003
	70	0.3342	0.0148		0.3342	0.0148	0.3343	0.0148	0.9974	0.0183	0.0005
	80	0.3530	0.0152		0.3530	0.0152	0.3530	0.0152	0.9912	0.0356	0.0020
Logarithmic	T (°C)	a			k		c		R^2^	MSD	χ^2^
	50	0.9773			0.0067		−0.0039		0.9993	0.0097	0.0001
	60	1.0270			0.0092		−0.0159		0.9988	0.0127	0.0002
	70	1.0142			0.0143		−0.0147		0.9978	0.0167	0.0003
	80	1.0851			0.0142		−0.0327		0.9929	0.0319	0.0012
Logistic	T (°C)	a_0_			a		k		R^2^	MSD	χ^2^
	50	0.1219			0.1251		0.0067		0.9993	0.0098	0.0001
	60	0.1489			0.1468		0.0096		0.9984	0.0147	0.0003
	70	0.1035			0.1032		0.0148		0.9974	0.0183	0.0004
	80	0.1153			0.1089		0.0152		0.9912	0.0356	0.0015
Midilli	T (°C)	a		k		n	b		R^2^	MSD	χ^2^
	50	0.9866		0.0088		0.9478	−0.000008		0.9995	0.0081	0.0001
	60	0.9907		0.0059		1.0963	−0.000008		0.9993	0.0100	0.0001
	70	0.9724		0.0086		1.1207	−0.000006		0.9986	0.0134	0.0002
	80	0.9852		0.0033		1.3463	−0.000007		0.9995	0.0089	0.0001
Newton	T (°C)	k							R^2^	MSD	χ^2^
	50	0.0071							0.9981	0.0155	0.0003
	60	0.0094							0.9981	0.0161	0.0003
	70	0.0148							0.9974	0.0184	0.0004
	80	0.0141							0.9867	0.0436	0.0020
Page	T (°C)	k				n			R^2^	MSD	χ^2^
	50	0.0100				0.9271			0.9993	0.0098	0.0001
	60	0.0066				1.0776			0.9991	0.0108	0.0001
	70	0.0114				1.0618			0.9981	0.0158	0.0003
	80	0.0039				1.3065			0.9992	0.0104	0.0001
Thompson	T (°C)	a				b			R^2^	MSD	χ^2^
	50	−35.1148				0.5118			0.9985	0.0138	0.0002
	60	−2247.7001				4.5874			0.9981	0.0161	0.0003
	70	−4120.3012				7.7995			0.9974	0.0184	0.0004
	80	−4282.2716				7.7620			0.9867	0.0436	0.0022
Verna	T (°C)	a			k		k1		R^2^	MSD	χ^2^
	50	−6.3939			0.0074		0.0074		0.9981	0.0156	0.0003
	60	−3.4090			0.0059		0.0066		0.9994	0.0087	0.0001
	70	−4.7678			0.0096		0.0104		0.9987	0.0128	0.0002
	80	−2.8776			0.0075		0.0089		0.9931	0.0315	0.0013

**Table 5 foods-12-02106-t005:** Effective diffusivities obtained from the drying kinetics of smelling and pout peppers at different temperatures.

Varieties	T (°C)	D_eff_ (m^2^ s^−1^)	R^2^
Smelling	50	5.75 × 10^−10^	0.9650
	60	7.34 × 10^−10^	0.9424
	70	10.62 × 10^−10^	0.9276
	80	15.16 × 10^−10^	0.9126
Pout	50	3.47 × 10^−10^	0.9753
	60	4.67 × 10^−10^	0.9646
	70	7.29 × 10^−10^	0.9663
	80	8.59 × 10^−10^	0.9386

**Table 6 foods-12-02106-t006:** Activation energy (E_a_) and coefficients of determination (R^2^) of smelling and pout peppers dried at temperatures of 50, 60, 70, and 80 °C.

Peppers	D_eff0_ (m^2^·s^−1^)	E_a_ (kJ·mol^−1^)	R^2^
Smelling	5.67 × 10^−5^	31.01	0.9875
Pout	2.56 × 10^−5^	30.11	0.9777

**Table 7 foods-12-02106-t007:** Thermodynamic properties of the drying kinetics of smelling and pout peppers dried at temperatures of 50, 60, 70, and 80 °C.

Peppers	T (°C)	ΔH (kJ·mol^−1^)	ΔS (kJ mol^−1^·K^−1^)	ΔG (kJ·mol^−1^)
Smelling	50	28.3243	−0.3269	133.9500
	60	28.2412	−0.3271	137.2199
	70	28.1581	−0.3274	140.4923
	80	28.0749	−0.3276	143.7671
Pout	50	27.4223	−0.3335	135.1890
	60	27.3392	−0.3337	138.5252
	70	27.2561	−0.3340	141.8638
	80	27.1729	−0.3342	145.2049

ΔH is enthalpy; ΔS is entropy; ΔG is Gibbs free energy.

**Table 8 foods-12-02106-t008:** Proximal composition of smelling and pout peppers in natura and powders obtained at different drying temperatures.

Parameters	Peppers	In Natura	Drying Temperatures (°C)			
			50	60	70	80
Water content (%)	Smelling	91.88 ± 0.16 aA	9.34 ± 0.14 bB	9.06 ± 0.03 bB	8.92 ± 0.05 aB	7.93 ± 0.04 aC
	Pout	83.30 ± 0.72 bA	11.95 ± 0.02 aB	10.41 ± 0.04 aC	7.25 ± 0.12 bD	6.56 ± 0.10 bE
Lipids (% d.b.)	Smelling	2.11 ± 0.20 aC	3.21 ± 0.08 bB	4.03 ± 0.27 bA	3.95 ± 0.07 bA	3.86 ± 0.03 bA
	Pout	2.10 ± 0.18 aD	6,02 ± 0.11 aB	4.58 ± 0.18 aC	5.78 ± 0.24 aB	7.70 ± 0.05 aA
Proteins (% d.b.)	Smelling	19.38 ± 0.58 aA	14.87 ± 0.30 aC	15.26 ± 0.21 aC	18.40 ± 0.15 aB	18.59 ± 0.35 aAB
	Pout	14.96 ± 0.39 bB	14.79 ± 0.43 aB	15.83 ± 0.18 aA	15.34 ± 0.27 bAB	15.39 ± 0.28 bAB
Ashes (% d.b.)	Smelling	6.83 ± 0.28 aA	5.60 ± 0.18 aC	6.30 ± 0.09 aB	6.73 ± 0.09 aAB	6.76 ± 0.14 aAB
	Pout	6.76 ± 0.47 aA	5.37 ± 0.22 aC	6.23 ± 0.06 aB	6.47 ± 0.07 aAB	6.21 ± 0.07 bB
Carbohydrates (% d.b.)	Smelling	71.67 ± 0.90 bC	76.32 ± 0.38 aA	74.41 ± 0.13 aB	70.92 ± 0.30 bC	70.80 ± 0.42 aC
	Pout	76.18 ± 0.88 aA	73.82 ± 0.20 bB	73.36 ± 0.34 bBC	72.41 ± 0.21 aC	70.71 ± 0.27 aD
Energetic value (kcal 100 g^−1^ d.b.)	Smelling	383.23 ± 1.59 aC	393.65 ± 1.06 bAB	394.97 ± 1.71 bA	392.83 ± 0.33 bAB	392.29 ± 0.71 bB
	Pout	383.43 ± 0.97 aE	408.63 ± 1.00 aB	398.00 ± 1.01 aD	403.04 ± 0.95 aC	413.65 ± 0.52 aA

Note: means followed by the same lowercase letter in columns and uppercase in rows do not differ statistically by Tukey’s test at 5% probability.

## Data Availability

The data presented in this study are available on request from the corresponding author.

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
