# Peer review of "Smelling Peppers and Pout Submitted to Convective Drying: Mathematical Modeling, Thermodynamic Properties and Proximal Composition"

_foods, 2023, doi:10.3390/foods12112106_

Round 1

Reviewer 1 Report

Pepper is wide scale used in the food production technologies. Dehydration preservation is one of the key issues in pepper processing. Detailed investigation of drying kinetics and effects of drying on microbial state and quality parameters can provide useful information for the practice. But, the drying kinetics of food and raw materials are invesstigated in numeroues papers. The manuscript focuses on the mathematical modelling of smelling peppers and Pout peppers,  proximal composition, thermodynamic properties and effective diffusivity. The manuscript contains interesting results, but, in my opinion, the novelty is low, and the Manuscript is too superficial.

Some comments, suggestions:

Please highlight the novelties of the study in the Introduction section.

Abstract is too general. The drying conditions are given, but results are not concretized (with absolute or relative changes of the parameters).

In the Introduction section, the (practical) relevance of thermodynamic properties and effective diffusivity should be discussed.

The humidity of drying air is not provided.

Please give how was the drying parameters selected/determined (temperature, velocity, time).

Please give references for the applied methods (drying kinetic models, effective diffusivity).

Please provide measuring errors/standard deviation for data presented in tables.

In lines 183-190 the air humidity of the drying experiments should be disccussed, as well.

Please reconsider the place of tables (table 3.4).

The visibility of Figure 1 and Figure 2 (mainly text and axis) is very low. Please improve it.

Table 8: Please consider to present data in figure(s) instead of table.

The Conclusion is too superficial, in my opinion.

Pepper is wide scale used in the food production technologies. Dehydration preservation is one of the key issues in pepper processing. Detailed investigation of drying kinetics and effects of drying on microbial state and quality parameters can provide useful information for the practice. But, the drying kinetics of food and raw materials are invesstigated in numeroues papers. The manuscript focuses on the mathematical modelling of smelling peppers and Pout peppers,  proximal composition, thermodynamic properties and effective diffusivity. The manuscript contains interesting results, but, in my opinion, the novelty is low, and the Manuscript is too superficial.

Some comments, suggestions:

Please highlight the novelties of the study in the Introduction section.

Abstract is too general. The drying conditions are given, but results are not concretized (with absolute or relative changes of the parameters).

In the Introduction section, the (practical) relevance of thermodynamic properties and effective diffusivity should be discussed.

The humidity of drying air is not provided.

Please give how was the drying parameters selected/determined (temperature, velocity, time).

Please give references for the applied methods (drying kinetic models, effective diffusivity).

Please provide measuring errors/standard deviation for data presented in tables.

In lines 183-190 the air humidity of the drying experiments should be disccussed, as well.

Please reconsider the place of tables (table 3.4).

The visibility of Figure 1 and Figure 2 (mainly text and axis) is very low. Please improve it.

Table 8: Please consider to present data in figure(s) instead of table.

The Conclusion is too superficial, in my opinion.

Author Response

Reviewer 1

The authors thank you for your attention in correcting the manuscript and accept the requested corrections. With exceptions justified by the authors throughout the file.

1. Please highlight the novelties of the study in the Introduction section.

It was added throughout the introduction.

2. Abstract is too general. The drying conditions are given, but results are not concretized (with absolute or relative changes of the parameters).

Information has been inserted throughout the summary.

3. In the Introduction section, the (practical) relevance of thermodynamic properties and effective diffusivity should be discussed.

It was inserted along the introduction.

4. The humidity of drying air is not provided.

The parameters were inserted in the topic “2.2 Drying kinetics”.

5. Please give how was the drying parameters selected/determined (temperature, velocity, time).

The information was inserted in the topic “2.2 Drying kinetics”

6. Please give references for the applied methods (drying kinetic models, effective diffusivity).

Information was provided in the Materials and Methods topic.

7. Please provide measuring errors/standard deviation for data presented in tables. In lines 183-190 the air humidity of the drying experiments should be disccussed, as well.

Data were entered and discussed.

8. Please reconsider the place of tables (table 3.4).

The authors believe that the location of the tables is appropriate for the sequence of discussion of the results, so we ask that you reconsider the request.

9. The visibility of Figure 1 and Figure 2 (mainly text and axis) is very low. Please improve it.

Figures have been edited and improved.

10. Table 8: Please consider to present data in figure(s) instead of table.

The authors believe that the amount of data present in the table would make it difficult to display it in the form of graphs, requesting that the request be reconsidered.

11. The Conclusion is too superficial, in my opinion.

Completion has been improved.

Reviewer 2 Report

This work is about the mathematical modeling, thermodynamic properties and proximal composition of convective drying of peppers. This work is well done, and could be accepted after revision.

1. The contents of Table 3 and Table 5 should be placed on the same page, or add a continuation table.

2. The value of R2 in the text should keep the same number of decimal places.

3. Table 6, the value of R2 decimal point is wrongly written, should be dotted instead of the tongsï¼›

4. Table 7, the decimal point of the value in the column of ΔS is wrongly written.

5, There are too many separate paragraphs in introduction. Suggest merging paragraphs appropriately. The first part of the Introduction of the first paragraph and the second paragraph content is similar, should be combined into one paragraph.

6. Page 2, line 87, the -1 after the unit in 1.0 m s-1 should be superscripted.

7. “as well as a reduction of 41.52 and 44.04%” 

45 of line 177 on page 5 should be written in full.

8. Is 183-195 on page 5 an analysis of the data in Table 1 or Table 2? I Suggest checking the content in the manuscript.

9. There are too many paragraphs in the manuscript, it is recommended to summarize them appropriately. For example, 3.4.

10. I suggest checking the grammar in the manuscript and correcting typing errors.

Author Response

Reviewer 2

The authors thank you for your attention in correcting the manuscript and accept the requested corrections. With exceptions justified by the authors throughout the file.

1. The contents of Table 3 and Table 5 should be placed on the same page, or add a continuation table.

As the presentation of results and discussions are topicalized, the authors believe that the sequence presented in the work does not allow this junction, since Table 3 and 4 refer to the data obtained in the mathematical modeling of the experimental data through the tested models, whereas Table 5 refers to the following stage of data evaluation, referring to the evaluation of the effective diffusivity in the process.

2. The value of R2in the text should keep the same number of decimal places.

The requested change was made throughout the entire manuscript.

3. Table 6, the value of R2decimal point is wrongly written, should be dotted instead of the tongsï¼›

Replacement was performed throughout the entire manuscript.

4. Table 7, the decimal point of the value in the column of ΔS is wrongly written.

The correction has been made.

5. There are too many separate paragraphs in introduction. Suggest merging paragraphs appropriately. The first part of the Introduction of the first paragraph and the second paragraph content is similar, should be combined into one paragraph.

The merge has been performed.

6. Page 2, line 87, the -1 after the unit in 1.0 m s-1should be superscripted.

The change has been made.

7. “as well as a reduction of 41.52 and 44.04%” 45 of line 177 on page 5 should be written in full.

The authors believe that the way in which the data were presented allows for greater understanding than written in full.

8. Is 183-195 on page 5 an analysis of the data in Table 1 or Table 2? I Suggest checking the content in the manuscript.

The discussions refer to Table 2, the typo has been corrected.

9. There are too many paragraphs in the manuscript, it is recommended to summarize them appropriately. For example, 3.4.

Changes were made throughout the manuscript.

10. I suggest checking the grammar in the manuscript and correcting typing errors.

Grammar has been corrected throughout the entire manuscript.

Reviewer 3 Report

Authors written an important topic relevant to present day industry and markets. This is interest to read and there are couple of  places need to be readdressed to increase the scientific content and clarify further for the interested readers. Also it will improve the quality of the manuscript.

1. I cannot see a photo of the experimental set up or schematic diagram of the experimental apparatus. This should be a must for the drying experiments as it will help readers and other researchers to adapt authors procedures. Please include those in the revise version of the manuscript.

2. Majority of countries pepper is drying using sun's heat. Need to describe it and compare with the advantages and disadvantages of the mechanical methods. Also effect of air velocity should be addressed?

3. Equation (1) uses RX as the moisture ratio. But in majority of the literature it is given as MR. So use of MR is appropriate to remove any misunderstanding of the readers.

4. Effective diffusivity is calculated using Fick's equation but it is not mentioned in the text. So authors need to use correct names and some references for the used equation. What approximations were used to equate the shape of the pepper. They mentioned the slices but I am confused what they mean by slices. A description or photo of slices should be included. Also how they solve the equation with nine terms was not described. Please include how you do it with what method in mathematics and any software used?

5. Fig 2 x-scale data is hard to read. Redraw the graphs to improve visibility.

6. This manuscript preparation involved so many authors but contribution of one author is minimum for an academic paper of this nature. This has to be addressed.

English should be checked again, it may be minor changes.

Author Response

Reviewer 3

The authors thank you for your attention in correcting the manuscript and accepted all requested corrections.

1. I cannot see a photo of the experimental set up or schematic diagram of the experimental apparatus. This should be a must for the drying experiments as it will help readers and other researchers to adapt authors procedures. Please include those in the revise version of the manuscript.

A graphical abstract has been added to the manuscript.

2. Majority of countries pepper is drying using sun's heat. Need to describe it and compare with the advantages and disadvantages of the mechanical methods. Also effect of air velocity should be addressed?

Information has been inserted in the Introduction and Materials and Methods.

3. Equation (1) uses RX as the moisture ratio. But in majority of the literature it is given as MR. So use of MR is appropriate to remove any misunderstanding of the readers.

The term has been replaced.

4. Effective diffusivity is calculated using Fick's equation but it is not mentioned in the text. So authors need to use correct names and some references for the used equation. What approximations were used to equate the shape of the pepper. They mentioned the slices but I am confused what they mean by slices. A description or photo of slices should be included. Also how they solve the equation with nine terms was not described. Please include how you do it with what method in mathematics and any software used?

The information has been entered in the Materials and Methods section.

5. Fig 2 x-scale data is hard to read. Redraw the graphs to improve visibility.

Figures in the manuscript were redone and improved.

6. This manuscript preparation involved so many authors but contribution of one author is minimum for an academic paper of this nature. This has to be addressed.

The authors' contributions were reassessed.

English should be checked again, it may be minor changes.

Language was checked throughout the entire manuscript.

Round 2

Reviewer 1 Report

The authors have revised the manuscript according to reviewers' comments and suggestions. Revision made the manuscript more complete and clear. I agree and accept all modifications made by the authors.